# Self-Care Practices of Primary Health Care Patients Diagnosed with Chronic Heart Failure: A Cross-Sectional Survey

**DOI:** 10.3390/ijerph16091625

**Published:** 2019-05-09

**Authors:** Rosalia Santesmases-Masana, Luis González-de Paz, Elvira Hernández-Martínez-Esparza, Belchin Kostov, Maria Dolors Navarro-Rubio

**Affiliations:** 1School of Nursing, Hospital Santa Creu i Sant Pau. Universitat Autònoma de Barcelona (UAB), 08025, Barcelona, Spain; RSantesmases@santpau.cat (R.S.-M.); elviraherna95@gmail.com (E.H.-M.-E.); 2Les Corts Primary Healthcare Center, Primary Healthcare Transversal Research Group, Institut d’Investigacions Biomèdiques August Pi i Sunyer (IDIBAPS), 08028 Barcelona, Spain; 3Primary Healthcare Transversal Research Group, Institut d’Investigacions Biomèdiques August Pi i Sunyer (IDIBAPS), 08028 Barcelona, Spain; badriyan@clinic.cat; 4Patient Experience Department, Hospital Sant Joan de Deu, Esplugues del Llobregat, Universitat Internacional de Catalunya, Sant Cugat del Vallés, 08950 Esplugues de Llobregat, Spain; mnavarror@sjdhospitalbarcelona.org

**Keywords:** heart failure, self-care practices, primary healthcare, health literacy

## Abstract

Chronic heart failure patients require self-care behaviors and active monitoring of signs and symptoms to prevent worsening. Most patients with this condition are attended in primary healthcare centers. This study aimed to evaluate the endorsement of and adherence to self-care behaviors in primary health care patients with chronic heart failure. We conducted a multicenter cross-sectional study. We randomly included chronic heart failure patients from 10 primary healthcare centers in the Barcelona metropolitan area (Spain). Patients completed the European Heart Failure Self-Care Behaviour Scale, a health literacy questionnaire. Differences between groups were studied using ANOVA tests. We included 318 patients with a mean age of 77.9 years, mild limitations in functional activity New York Heart Association scale (NYHA) II = 51.25%), and a low health literacy index of 79.6%. The endorsement of self-care behaviors was low in daily weighing (10.66%), contacting clinicians if the body weight increased (22.57%), and doing physical exercise regularly (35.58%). Patients with lower educational levels and a worse health literacy had a lower endorsement. The screening of individual self-care practices in heart failure patients might improve the clinician follow-up. We suggest that primary healthcare clinicians should routinely screen self-care behaviors to identify patients requiring a closer follow-up and to design and adapt rehabilitation programs to improve self-care.

## 1. Introduction

Heart failure is defined as “a structural and/or functional cardiac abnormality resulting in reduced cardiac output and/or elevated intracardiac pressures at rest or during stress” [1]. Heart failure affects the quality of life and the activities of daily living and reduces life expectancy [2]. The severity of heart failure is based on a scale of limitations in the functional physical activity [1]. Heart failure is considered stable when the symptoms and signs are unchanged for ≥1 month. Heart failure tends to worsen, and hospital admissions are frequent [1]. Around 26 million adults have heart failure worldwide [3]. The prevalence is 2% in developed countries and rises to 8% in patients aged >75 years. In countries with aging populations, heart failure is becoming a public health problem [4]. 

Patients with heart failure require specific healthcare that can increase life expectancy and the quality of life. Healthcare services such as post-hospital discharge follow-up, specific care plans, and education and monitoring programs can improve the prognosis [5]. In countries with Primary Health Care (PHC) systems, such as Spain, patients with stable heart failure are attended life-long in PHC centers by physicians and family nurses. When PHC clinicians suspect a worsened heart function, patients may be referred to the hospital. This easy access to health care professionals, compared with hospital-based systems, has been shown to be more cost-efficient [6]. 

PHC clinicians play a key role in identifying patients with a lack of or poor skills in managing heart failure. Cognitive and functional limitations, old age, low educational levels and inadequate social support are adherence-related factors that hamper adequate self-care [7]. PHC planned care has been shown to reduce heart failure re-hospitalizations and maintain the patient quality of life [2,8]. When patients are admitted to a hospital, the PHC receives an automatic warning, and before the discharge the hospital sends a report to the PHC, to the nurse and to the family physician. They carry out a phone call to appoint a visit at home or at the PHC center [9,10]. While patient care in the PHC system is lifelong, patients with a recent hospital admission receive higher attention to set a new plan of care. PHC clinicians’ objectives in heart failure patients includes activities of self-management to improve patient decision-making, such as education on the self-detection of symptoms if heart failure worsens (e.g., weight control to detect fluid retention), control of adherence to prescribed drugs, diet and exercise routines, and symptom management [11,12]. Clinical guidelines on heart failure recommend drawing up follow-up procedures and standardizing interventions; however, the self-care of heart failure patients strongly depends on individual factors that PHC physicians and family nurses must take into account [13]. Age and cognitive impairment result in a loss of adherence to treatment, and elderly patients are often confused about the early recognition of symptoms [14]. Anxiety and depression in patients with heart failure are associated with less physical activity, unhealthier diets and reduced treatment adherence [15]. Social support has been shown to reduce avoidable hospital admissions and better self-care overall [7]. Health literacy is the knowledge, skills and health-related experience that allows patients to recognize their health condition and how to manage their own health care [16]. An inadequate health literacy has been shown to raise morbidity and mortality and to worsen self-care [17,18]. Self-perception with respect to mistaking symptoms or signs of heart failure often occurs in patients with concomitant diseases, such as chronic renal failure and diabetes [19]. Self-care follow up is associated with a better health status in heart failure patients [20,21]. Most studies have examined patients with unstable or acute heart failure attended in hospitals or hospital outpatient clinics during the clinical follow-up [22] and have found average levels of self-care and a low adherence to each self-care behavior. However, most patients who have a stable heart failure are autonomous and are routinely seen in PHC centers by family nurses and physicians, but their endorsement of self-care behaviors and their socio-demographic and clinical determinants have not been described. The aim of the study was to evaluate the endorsement of and adherence to self-care behaviors in primary health care patients with a chronic heart failure.

## 2. Materials and Methods 

### 2.1. Aim of the Study

To determine the endorsement of self-care behaviors in PHC patients diagnosed with chronic heart failure, and to examine potential relationships between self-care practices and sociodemographic and clinical characteristics. 

### 2.2. Study Design

We made a cross-sectional survey in Barcelona, Spain. The questionnaire used to measure the main variable was studied through a psychometric analysis.

### 2.3. Participants and Sample

We included patients with a documented chronic heart failure from all 10 PHC centers of Hospitalet de Llobregat [23], a city in the Barcelona metropolitan area which, in 2014, attended to 215,816 patients [24]. To maximize participation, we included patients aged ≥55 years, as this age group has a higher prevalence of heart failure [25]. The inclusion criteria were: ≥1 PHC visit during 2014 and the ability to attend a PHC visit. The exclusion criteria were: having documented cognitive impairment (e.g., Alzheimer or dementia), severe dependence (Barthel Index < 20), life expectancy < 1-year, severe mental disease, and not being home-dwelling. The medical records of the 10 PHC centers showed 2762 patients registered with a diagnosis of heart failure. We randomly sampled patients using replacements to maintain a proportionality between the PHC centers. A previous study showed that almost 50% of patients with heart failure attended to in hospitals had a moderate level of health care behavior [26], and therefore the minimum sample size that was needed was calculated as 298 patients, with 95% confidence intervals and a precision of ±6% units, as well as an anticipated replacement rate of 20%.

### 2.4. Data Collection

Ten PHC nurses contacted patients fulfilling the inclusion criteria by telephone, provided them with information on the study aims, and invited them to participate. If patients accepted, they were given a PHC center appointment. Data collection took place between January and May 2015. A pilot study, including 31 patients fulfilling the inclusion criteria, was carried out before starting the data collection, to verify that all researchers followed the research protocol. 

### 2.5. The Self-Care Questionnaire and Other Variables

The European Heart Failure Self-Care Behavior Scale (EHFScBS) is a self-administered questionnaire consisting of 12 items on self-care skills and attitudes that allow the monitoring of the health-related behavior of heart failure patients [27]. Each item is rated on a scale of 5 points, from 1 “I don’t agree at all” to 5 “I completely agree”. Scores range from 12 to 60, and higher scores reflect poorer self-care. The content validity of the EHFScBS was verified by a concept analysis, expert panels and face-validity. Psychometric properties were validated using a concurrent validity assessment, and the internal consistency showed a Cronbachs’ alpha of 0.81. The EHFScBS has been widely used in research as it is a unique instrument [27] and has been adapted into Swedish, Dutch, Finnish, Italian, Turkish and Spanish [26]. 

We collected sociodemographic variables: age, sex, marital status, number of cohabitants, educational level, income and source of income. The health literacy was measured using the Health Literacy Survey-European Union-Q47 (HLS-EU-Q47) questionnaire [28], which includes 47 items related to the management of health information, and aspects of healthcare, disease prevention and health promotion. The HLS-EU-Q47 in Spanish has a unidimensional structure with a high internal consistency (Cronbach’s α = 0.97) [29]. The questionnaire provides a general index of health literacy, and results might be expressed in a standardized form from 0 to 50; furthermore, the authors defined four levels of health literacy in the general public: inadequate, problematic, sufficient, and excellent. Other clinical characteristics collected were: the severity of the heart failure, assessed with the New York Heart Association scale (NYHA) [30], polypharmacy (>5 prescribed drugs), the adherence to drug treatment, assessed using the Morisky-Green test [31], and the number of PHC nurse and family physician visits in the last year.

### 2.6. Data Analysis

#### 2.6.1. Rasch Analysis

The EHFScBS questionnaire was analyzed using a Rasch probabilistic model. The Rasch analysis allows ordinal measures to be transformed into intervals (log-odds units or logit), and all EHFScBS items to be arrayed individually from a higher to lower endorsement; it also allows patients to show worse to better adherence to self-care. The Rasch analysis requires the goodness-of-fit of data to the model to be examined: if the assumptions are satisfied, this provides researchers with evidence of the appropriate psychometric properties and validity of the questionnaire [32]. 

#### 2.6.2. Rasch Model Selection and Scale Structure Functioning

A Rasch model was used to rate scales for polytomous responses (RSM) because all items shared an identical rating scale structure. After the analysis, the rating scale structure was examined: each category must contribute with substantive meaning to the rating scale functioning. Thresholds between the categories of each item—the point where there is a fifty percent probability of a response falling in one or another adjacent category—should be ordered and distinguishable. There may be disordering when questionnaires contain too many category options or if a category is incorrectly labeled. This was detected by an analysis of frequencies, thresholds and item category curves (ICC) for each item. The categories were collapsed according to the following guidelines [33]: (1) Each category should contain ≥10 observations; (2) The average measures must be ordered; and (3) The thresholds must be ordered. Once the structure functioning was verified, the reliability was evaluated according to the Item Separation Index (comparable to Cronbach’s α) [32]: values ≥ 0.7 show the that the scale is differentiated between individuals and items along the latent trait.

#### 2.6.3. Item Fit Analysis, Local Independence and Unidimensionality

The goodness-of-fit of each item to the Rasch model was studied using mean square fit statistics: Infit/Outfit, with mean-square values from 0.7 to 1.4 being expected [34]. Local independence (item dependence) assumes that responses to items are not related, and that they therefore contribute independently to the construct of the questionnaire. This was examined by identifying positive correlations among the item residuals exceeding ±0.3. Unidimensionality assumes that items are measurable along a single trait continuum. Differences at the item level were calculated using a paired t-test utilizing two subsets of items identified by a principal components analysis of the residuals (PCAR). The percentage of tests outside the 95% confidence intervals was expected not to exceed 5%. If no significance was found at the item level, the scale was classified as unidimensional [32,33,34]. 

#### 2.6.4. Targeting and Group Differences

The targeting of the patients’ self-care according to the relative endorsement for each item was made utilizing the figure of the Wright map, where the left side shows the self-care adherence of each patient ordered from worse (lower) to better (higher). The right-side orders items from a higher (inferior level) to lower (superior level) probability of endorsement. If an item and a patient have the same position, a 50% likelihood of endorsement of the item can be expected. As the patient’s self-care adherence and item endorsement are on an identical logit scale, this allows a visual comparison of the items. Between-group differences in overall self-care were tested by ANOVA for the overall person mean location by groups of: sex, age (groups according to quartiles), marital status, cohabitation, academic level, NYHA classification, adherence to drug treatment, polypharmacy, hospital admission last year and health literacy index (inadequate, problematic, sufficient, and excellent), and Pearson’s correlation. Significant variations or correlations in the mean groups indicated a differing adherence to self-care practices between the groups. In groups with significant differences we analyzed the mean difference with the Bonferroni pairwise comparison test; logits measures were transformed into odds ratios (OR) with the 95% CI.

### 2.7. Ethics Approval and Consent to Participate

The Jordi Gol Institute of Primary Care Research Ethics Committee approved the study (Ref. number: P14/035). All participants gave written and verbal informed consent to participate. The researchers complied with the Declaration of Helsinki directives, and all the local laws concerning biomedical studies, data protection and respect for human rights. In the analysis, all data were handled anonymously.

## 3. Results

### 3.1. Characteristics of Participants

We contacted 335 patients, of which 318 (94.9%) agreed to participate. The main reason for nonparticipation was an unwillingness to attend an extra PHC appointment. The mean age was 77.96 years. About 58% had not completed primary education, the majority were NYHA II (51.3%) and non-adherence to a drug treatment was 75.5%. The average health literacy index was 25.4 logits, indicating a problematic or lower health literacy in 79.6% of participants. The mean age for women was significantly higher (mean age for males = 75.97 vs. mean age of woman = 79.85, mean difference 3.88 years, *t* = −4.086; *p* < 0.001) (Table 1).

### 3.2. Rasch Analysis of EHFScBS Questionnaire

The Rasch model analyses showed that item category curves (ICC) had reversed thresholds in all items, indicating problems with the categorization. Figure 1 shows the rating scale thresholds with the five-point rating scales: categories 2, 3, and 4 were shadowed by the lower and higher categories, suggesting a dichotomous pattern (e.g., Yes and No). The inspection of the frequency of scores in Table 2 showed that three items (7, 10 and 11) had some categories with a frequency <10, and in item 7 the average measures could not be ordered between the third and fourth categories. Therefore, the subsequent analyses collapsed one category at a time for all items, according to the name of each category. However, the ICC plots did not improve until a dichotomous pattern was formed. In the final analysis, categories 1 and 2 were collapsed, as were categories 3 to 5. This fact did not affect the robustness of the estimate but improved the Rasch reliability to 0.99.

The item goodness-of-fit statistics (Outfit and Infit) (Table 3) show that all items were in the range of 0.71–1.47, except for item 10 with an Outfit of 0.51: this indicated that participants with a poor or higher endorsement had somewhat predictable values—the Guttman pattern—and we decided to keep this item after checking that it did not alter the stability of the estimates or other analyses. The residual correlations were all ≤0.3, meaning that the local dependence was not substantive. The independent *t*-test, comparing participants’ measures from the positive and negative PCAR loadings, was 2.51% (*n* = 8), showing that the items were aligned in one latent trait continuum, meaning a single latent trait was measured without a substantive secondary dimension.

Item 12 (“I exercise regularly”) had a Differential Item Functioning (DIF) between males and females (Mantel-Haenszel = 12.235, *p* < 0.001); males had a better mean endorsement than females of −1.01 logits. This DIF was expected because females were, on average, older than males, and older age can affect physical activity. We solved the DIF by splitting the item and calibrating a measure of adherence separately for each group; this did not affect any other item and improved the robustness of the item estimate. After splitting the item, the separation index (analogous to the Cronbach alpha) was 0.99, indicating an optimal reliability.

In the Rasch analysis, the overall mean endorsement of the EHFScBS was −2.23 logits. The items are ordered from a lower to higher endorsement. The lowest adherence of the majority of participants was for item 1 (“I weigh myself every day”), item 5 (“If I gain 2 kg in 1 week, I contact my doctor or nurse”) and the split item 12 for females (“I exercise regularly”). The item with the highest likelihood of endorsement was, “I take my medication as prescribed”, where almost all participants reported a high adherence. The Wright map (Figure 2) shows these results graphically. The mean endorsement of items (marked with a red “M” in the map) is below the mean of the self-care of patients, showing that the adherence to self-care was notable: a total of 34 (10.7%) patients showed the highest self-care because they surpassed the endorsement required for all items, and 7 items (4, 3, 9, 11, 2, 7, and 10) had a likelihood of endorsement of >90% (Table 3).

### 3.3. Targeting and Group Differences

We found significant between-group differences in the mean self-care adherence by education and by the health literacy index (Table 4). The pairwise comparison of these groups showed, in Table 5, that the OR of adherence to self-care in the group with a secondary education or higher, compared with patients who had not completed a primary education, was protective; furthermore, the adherence to self-care was also protective in the group with a sufficient health literacy, compared to the group with an inadequate health literacy. No correlation was observed between the number of PHC nurse visits (*r* = −0.24, *p* = 0.670) and PHC family physician visits (*r* = 0.03, *p* = 0.585).

## 4. Discussion

We studied the endorsement of self-care behaviors in PHC patients with chronic heart failure. Most patients received only PHC care and were not greatly affected by heart failure. Behaviors requiring shared responsibility and decision making to manage heart failure had a lower endorsement, and behaviors related to avoiding physical fatigue had a greater endorsement. Factors such as the educational level and health literacy were associated with a better adherence to treatment.

The analysis of the goodness-of-fit of data to the Rasch model of the EHFScBS did not change the questionnaire itself, but provided insights into the rating scale functioning of the questionnaire. The sample of patients answered the questionnaire in a clear dichotomous pattern, suggesting that the 5-category rating scale could be changed to a simple Yes or No answer. To our knowledge, neither the original validation study, nor any subsequent studies using the EHFScBS, have commented on this aspect [20,26,27,35,36]. As far as we know, this is the first analysis of the EHFScBS using a Rasch model; all other studies used classical test theory methods where the measure of self-care is the sum of scores, and therefore more categories of the rating scale are useful but they add more variability. However, using the Rasch model avoids this issue, and also provides researchers with results at an item level [37]. We believe that a Rasch model analysis might help clinicians to monitor patients’ attitudes, and the results could allow physicians and family nurses to answer the question “what should we do next?” more accurately, as well as personalize healthcare plans with questionnaires used in daily practice [38,39]. 

The behavior with the greatest endorsement in the EHFScBS questionnaire was related to the adherence to a drug treatment, similar to the results of a study including heart failure patients from 15 countries, including Spain. However, the participants were younger, (mean age = 69 years vs. 77.9 in our study), and were treated in outpatient clinics and hospitals [26,35]. According to the Morisky Green test, only 25% of patients adhered to a drug treatment. We believe that the item related to drug treatment in the EHFScBS has low sensitivity compared to the Morisky Green test. Adherence to a drug treatment is key in the management of chronic heart failure. PHC and home planned care support patients in organizing a medication schedule, e.g., arranging day and night time doses, updating on-line drug prescriptions and helping family members or home caregivers to manage different situations [7,40]. Our study suggests that the greater likelihood of an endorsement of adherence to a drug treatment might be linked to the high percentage of patients who were living with a partner, as they acted as informal caregivers. Frequent PHC nurse visits might also increase treatment adherence [41]. PHC providers should emphasize this point at each visit, clarifying any new or changed medications to patients and caregivers during all visits [42]. Other behaviors with a high endorsement were related to resting when feeling fatigue symptoms (e.g., “I take a rest during the day” or “If I am short of breath, I take it easy”). However, we think that these behaviors are not strange because physical impairment impels patients to change attitudes toward any issue or daily activity. Items with a lower endorsement were related to daily weighing and contacting clinicians over increases in body weight. Weighting is strongly recommended in heart failure guidelines, because of being used to change the dosage of diuretics and to alert physicians to a sudden worsening in the health status [1,11]. Other studies have shown a low adherence to this basic self-care behavior [26,35], and it might be regarded as a process of adaptation: patients may tolerate edema, weight gain, and fatigue for as long as seven days before seeking healthcare attention [43]. We think that evidence of a low adherence to weighing might alert health care professionals into implementing new strategies to help patients.

No correlation was found between the number of visits to PHC physicians and PHC nurses and the adherence to self-care; addressing a low adherence in patients with a good functional status that does not require frequent visits to PHC professionals would require initiating specific self-care programs soon after the heart failure diagnosis. Thereafter, PHC nurses could measure self-care behaviors, reinforcing positive behaviors, and identifying educational opportunities. However, this standardized health care model is not used in current clinical practice [44]. Our results suggest that health care providers might seek effective measures to help PHC professionals to follow up patients with heart failure.

We found that a majority of women had a lower endorsement of regular physical exercise than men. Women more often abandon physical exercise due to non-medical reasons, such as family commitments, and often have less time for exercise because of a multiplicity of roles (work, family, community, etc.) [45]. An adequate adherence to physical exercise programs remains a challenge in heart failure patients. In general, physical activity programs fail to address adherence barriers prior to the exercise initiation, or they improve adherence while follow-up is scarce [46]. A Spanish study estimated that only 6.9% of heart failure patients were included in a heart rehabilitation program [47] lower than the 20–30% found in other European countries. Two reasons for the scarcity of heart failure rehabilitation and physical exercise programs are: (1) the opinions of most health care professionals, who are much more oriented toward drug treatment than other health care interventions, and (2) difficulties in accessing programs, most of which take place in urban hospitals or far from patients’ households, which is an important barrier for the elderly [48,49]. We believe that designing PHC heart failure rehabilitation programs could increase the adherence to physical exercise and might improve self-care due to the proximity of PHC centers to patients’ home and to the multidisciplinary intervention of physicians and family nurses, which could be coordinated with specialist heart physicians [1,50,51]. 

The mean health literacy of patients was inadequate. This, together with the educational level, is reported to be associated with a greater morbidity and mortality and a lower treatment adherence [17,52]. Patients with heart failure and poor health literacy had difficulties navigating the health system and understanding the information required for self-care management [20,53]. The importance of health literacy and its correlation with health care [20,54] suggests that PHC clinicians should identify patients with low health literacy or educational levels. Once detected, nurses could adapt appointments and heart failure follow-up to target these factors. A possible problem is the lack of instruments to measure health literacy skills in daily clinical practice. However, PHC clinicians must be conscious of patients who may not understand healthcare messages, including the elderly, those with low educational levels and those lacking social support [7,55]. PHC physicians and family nurses should ensure that patients with inadequate health literacy understand written and oral information on self-care [56,57,58,59]. 

### 4.1. Limitations

Our study had some limitations. First, the HLS-EU is a general questionnaire used to examine health literacy in the general public, rather than patients with specific healthcare needs. However, as there are no appropriate instruments adapted into Spanish that measure health literacy in heart failure patients, we suggest it is necessary to utilize this questionnaire. Second, we only studied home-dwelling patients, thus excluding residents of nursing homes who are also attended by PHC staff, and this could bias the results. However, we aimed to measure only highly-autonomous patients and, in Spain, most elderly nursing home residents are disabled or have a low autonomy. Third, associations could not be studied due to our cross-sectional design. However, we believe that our results accurately describe self-care behaviors in heart failure patients and that they might aid PHC managers and researchers to develop health strategies and policies that improve these patients’ situations. Furthermore, longitudinal studies evaluating the effectiveness of programs in improving self-care behaviors are necessary.

### 4.2. Implications

Self-care behaviors have been evaluated in patients with acute heart failure attended to by hospitals or routinely by outpatient clinics, but most patients with chronic heart failure are attended to by primary healthcare family physicians and family nurses. The individual adherence to self-care behaviors has not been studied before. The results of this study will help develop procedures to improve the healthcare offered by nurses to these patients and to improve their self-managed behavior.

Self-care behaviors requiring the monitoring of the physical status and decision-making had a lower endorsement, and behaviors related to avoiding physical fatigue had a greater endorsement. The adherence to physical exercise differed between men and women. The educational level and health literacy were associated with an adherence to the treatment regime. Primary healthcare clinicians could improve self-care behavior screening and not only drug prescription regimes. Primary healthcare rehabilitation programs for patients with chronic heart disease could, according to their needs and behaviors, increase the adherence to health-related habits. Patients with a low level of health literacy should be identified to ensure a closer supervision of their self-care practices and to adapt educational messages to their specific needs.

## 5. Conclusions

Self-care behaviors of PHC patients with chronic heart failure were not optimal. PHC clinicians should improve the screening of self-care behaviors, including daily weighing and regular physical exercise. Patients with a low health literacy should be identified, and their self-care behavior should be supervised more closely to ensure the patient’s understanding and acceptance of educational messages. Designing and adapting heart rehabilitation programs in PHC could improve the monitoring of healthcare behaviors in heart failure. An appropriate long-term control of self-care behaviors could improve the equity of the healthcare provision.

## Figures and Tables

**Figure 1 ijerph-16-01625-f001:**
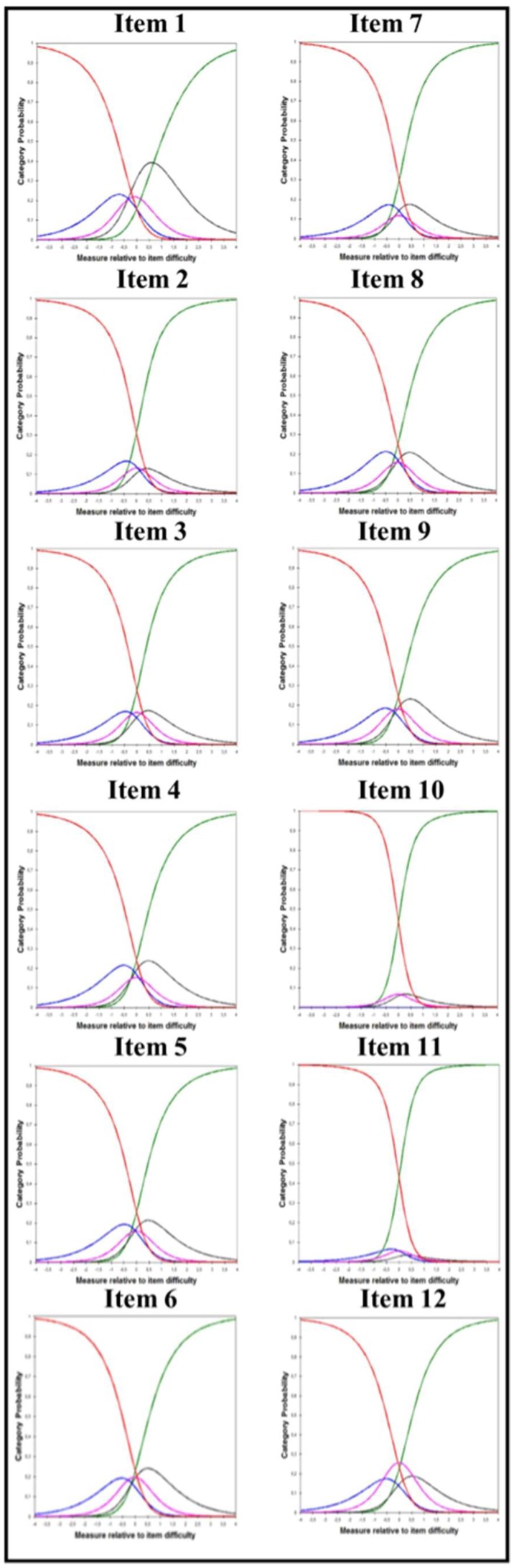
Item category curves (ICC) for the 12 items. All items showed disordered thresholds that suggested that the categories should be collapsed.

**Figure 2 ijerph-16-01625-f002:**
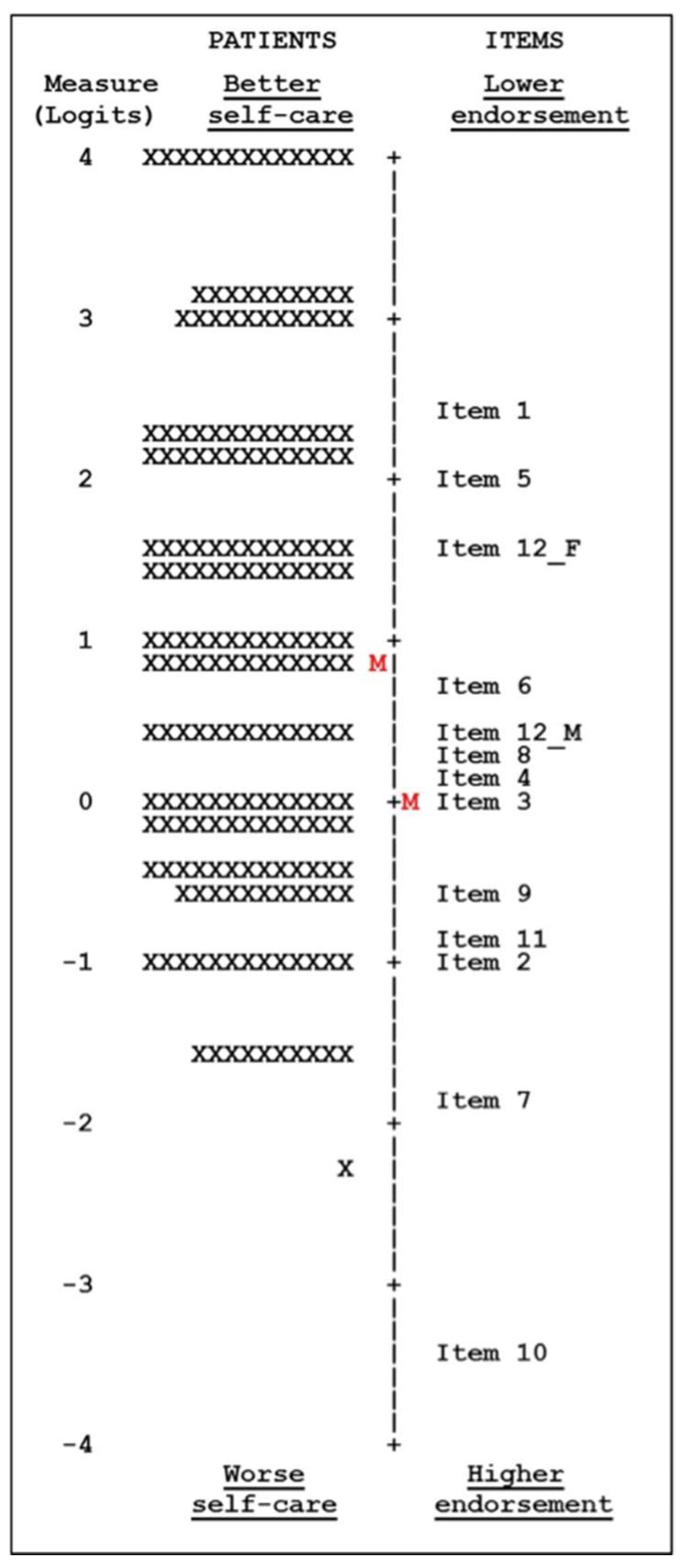
Map of items and persons. Each “X” represents 1 patient. Self-care and endorsement of items are measured in the same units. The “M’s” represents the mean level of self-care on the patients’ side and the endorsement location on the items’ side**.**

**Table 1 ijerph-16-01625-t001:** Participant characteristics. New York Heart Association (NYHA) Functional Classification.

	*N* = 318	Females	Males
**Sex *n* (%)**		163 (51.26%)	155 (48.74%)
**Age Mean (SD)**	77.96 (8.67)	78.85 (7.68)	75.97 (9.23)
**Marital status *n* (%)**			
Single	25 (7.86%)	13 (7.98%)	12 (7.74%)
Married or living together	180 (56.60%)	59 (36.20%)	121 (78.06)
Divorced	4 (1.26%)	1 (0.61%)	3 (1.94%)
Widowed	109 (34.28%)	90 (55.21%)	19 (12.26%)
**Cohabitation *n* (%)**			
Living alone	79 (24.84%)	62 (38.04%)	17 (10.97%)
2	185 (58.18%)	79 (48.47%)	106 (68.39%)
>2	54 (16.98%)	22 (13.50%)	32 (20.65%)
**Sources of household income *n* (%)**			
Employment / unemployment allowance	15 (4.72%)	6 (3.68%)	9 (5.81%)
Disability pension	21 (6.60%)	3 (1.84%)	18 (11.61%)
Retirement pension	174 (54.72%)	51 (31.29%)	123 (79.35%)
Widow’s pension	64 (20.13%)	62 (38.04%)	2 (1.29%)
Social aid (welfare financial benefits)	14 (4.40%)	13 (7.98%)	1 (0.65%
**Academic level *n* (%)**			
Primary education not completed	185 (58.18%)	117 (71.78%)	68 (43.87%)
Primary education	76 (23.89%)	34 (20.86%)	42 (27.10%)
Secondary Education/Vocational studies.	52 (16.35%)	12 (7.36%)	40 (25.81%)
University degree or higher.	5 (1.57%)	0 (0.00%)	5 (3.23%)
**NYHA *n* (%)**			
I	88 (27.67%)	43 (26.38%)	45 (26.38%)
II	163 (51. 25%)	83 (50.92%)	80 (51.61%)
III	64 (20.13%)	35 (21.47%)	29 (18.71%)
IV	3 (0.94%)	2 (1.23%)	1 (0.65%)
**Non-adherence to drug treatment *n* (%)**	240 (75.5%)	125 (76.69%)	115 (74.19%)
**Health Literacy Index (min: 0, max: 50). Mean (SD)**	25.44 (9.05	23.44 (9.22)	27.54 (8.41)
**Visits to PHC nurse in last year. Mean (SD)**	12.48 (11.02)	13.77 (11.90)	11.22 (9.90)
**Visits to PHC family physician in last year** **.** **Mean (SD)**	10.19 (6.30)	10.77 (11.90)	9.59 (5.75)

**Table 2 ijerph-16-01625-t002:** Analysis of the rating scale structure of the EHFScBS questionnaire. Items and frequency (%) of answers in each category. Rasch analysis requires categories with ≥10 observations. (In bold, categories with <10 of responses).

Item	Category Frequency (% of Responses)				
	1	2	3	4	5
	I Don’t Agree at all				I Completely Agree
1. I weigh myself every day.	129 (40.6%)	63 (19.8%)	50 (15.7%)	50 (15.7%)	26 (8.2%)
2. If I get short of breath, I take it easy.	23 (7.2%)	15 (4.7%)	23 (7.2%)	34 (10.7%)	223 (70.1%)
3. If my shortness of breath increases, I contact my doctor or nurse.	45 (14.2%)	25 (7.9%)	38 (11.9%)	47 (14.8%)	163 (51.3%)
4. If my feet/legs become more swollen than usual, I contact my doctor or nurse.	45 (14.2%)	34 (10.7%)	37 (11.6%)	64 (20.1%)	138 (43.4%)
5. If I gain 2 kg in 1 week, I contact my doctor or nurse.	128 (40.3%)	50 (15.7%)	39 (12.3%)	37 (11.6%)	64 (20.1%)
6. I limit the amount of fluids I drink (not more than 1.5–2 one/day).	59 (18.6%)	36 (11.3%)	51 (16.0%)	61 (19.2%)	111 (34.9%)
**7. I take a rest during the day.**	**9 (2.8%)**	**8 (2.5%)**	14 (4.4%)	41 (12.9%)	246 (77.4%)
8. If I experience increased fatigue, I contact my doctor or nurse.	51 (16.0%)	36 (11.3%)	39 (12.3%)	55 (17.3%)	137 (43.1%)
9. I eat a low salt diet.	22 (6.9%)	18 (5.7%)	35 (11.0%)	64 (20.1%)	179 (56.3%)
**10. I take my medication as prescribed.**	**4 (1.3%)**	**0 (0.0%)**	**4 (1.3%)**	11 (3.5%)	299 (94.0%)
**11. I get a flu shot every year.**	48 (15.1%)	**8 (2.5%)**	11 (3.5%)	**9 (2.8%)**	242 (76.1%)
12. I exercise regularly. Females	63 (19.8%)	34 (10.7%)	67 (21.1%)	46 (14.5%)	108 (34.0%)

**Table 3 ijerph-16-01625-t003:** Results of the Rasch model analysis.

Item	Measure (Logits)	Standard Error	Likelihood (%) of Endorsement of Each Item	Goodness of Fit Statistics		PCAR *
				Infit	Outfit	
1. I weigh myself every day.	2.48	0.16	10.66	0.98	0.99	−0.17
5. If I gain 2 kg in 1 week, I contact my doctor or nurse.	1.94	0.14	22.57	0.83	0.85	0.4
12F. I exercise regularly. Females	1.54	0.19	35.58 (females)	1.18	1.25	−0.14
6. I limit the amount of fluids I drink (not more than 1.5–2 one/day).	0.68	0.13	55.48	1.18	1.23	−0.3
12M. I exercise regularly. Males	0.4	0.19	60.00 (males)	1.18	1.29	−0.2
8. If I experience increased fatigue, I contact my doctor or nurse.	0.35	0.13	69.27	0.77	0.71	0.71
4. If my feet/legs become more swollen than usual, I contact my doctor or nurse.	0.17	0.13	91.22	0.89	0.88	0.53
3. If my shortness of breath increases, I contact my doctor or nurse.	0.03	0.13		0.93	0.94	0.54
9. I eat a low salt diet.	−0.61	0.15		1.06	1.08	−0.41
11. I get a flu shot every year.	−0.79	0.15		1.12	1.37	−0.3
2. If I get short of breath, I take it easy.	−0.93	0.16	99.37	0.99	1.00	−0.2
7. I take a rest during the day.	−1.85	0.2		0.98	1.47	−0.28
10. I take my medication as prescribed.	−3.4	0.37	99.69	0.99	0.51	−0.31

* PCAR: Loadings from principal component analysis of residuals of the Rasch model.

**Table 4 ijerph-16-01625-t004:** Differences between groups in mean self-care practices (ANOVA test).

Group	Sub-Group	*n*	Mean (SD) Self-Care	Statistic (*F*)	*p*-Value
Sex	Female	163	0.85 (1.40)	1.737	0.189
	Male	155	1.05 (0.13)		
Age	<72 years	78	0.99 (1.44)	0.459	0.711
	73 to 78 years	80	1.05 (1.30)		
	79 to 88 years	80	0.94 (1.26)		
	>89 years	80	0.81 (1.39)		
Marital Status	Single	25	0.88 (1.31)	1.477	0.221
	Married or living together	180	1.04 (1.38)		
	Widowed	109	0.77 (1.26)		
	Divorced	4	1.82 (1.95)		
Cohabitation	Alone	79	0.94 (1.20)	0.832	0.436
	2 residents	185	1.01 (1.44)		
	>2 residents	54	0.74 (0.74)		
Academic level	Primary education not completed	185	0.89 (1.31)	4.38	0.013
	Primary education completed	76	0.75 (1.30)		
	Secondary education or higher	57	1.40 (1.44)		
NYHA	I	88	1.15 (1.42)	1.06	0.365
	II	163	0.86 (1.37)		
	III	64	0.91 (1.17)		
	IV	3	0.47 (0.49)		
Adherence to drug treatment	Yes	78	0.98 (1.22)	0.069	0.793
	No	240	0.94 (1.39)		
Polypharmacy	Yes	272	0.99 (1.32)	2.058	0.152
	No	46	0.68 (1.45)		
Hospital admission last year	Yes	247	0.94 (1.38)	0	0.938
	No	71	0.95 (1.19)		
Health literacy index	Inadequate	147	0.63 (1.21)	6.75	0.001
	Problematic	106	0.98 (1.19)		
	Sufficient	57	1.67 (1.67)		
	Excellent	8	1.28 (0.97)		

**Table 5 ijerph-16-01625-t005:** Pairwise comparison of the means (Bonferroni method).

Group	Sub-Group	OR (95% CI)
Academic level	Primary education not completed *	
	Primary education completed	1.15 (0.74 to 1.78)
	Secondary education or higher	0.60 (0.37 to 0.97)
Health literacy Index	Inadequate *	
	Problematic	0.70 (0.45 to 1.09)
	Sufficient	0.35 (0.21 to 0.60)
	Excellent	0.52 (0.15 to 1.83)

The logit measure is expressed in OR. * is the reference group.

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
