# Peer review of "Self-Care Practices of Primary Health Care Patients Diagnosed with Chronic Heart Failure: A Cross-Sectional Survey"

_ijerph, 2019, doi:10.3390/ijerph16091625_

Round 1

Reviewer 1 Report

General

Do not use acronyms that are not universally accepted eg. HF - write in full

Material and Methods

Provide details of psychometrics of two tools

NYHA - please write in full first time used

Specify groups which will be compared in data analysis section

Results

Please report data consistently eg. mean (sd) and n (%)

I would suggest doing Table 1 by Gender

Include in Table 1 the total possible score for HLI etc to provide context for reader

Table 2 - specify details of items

Section 3.3 - It would have been good to see the table of OR (CI) in a table

Why was correlation of PHC visit and PHC physician visit done? 

Please add the psychometrics (Chronbach Alpha) as stated in methodology in the results

Author Response

Reviewer 1.

General

1. Do not use acronyms that are not universally accepted eg. HF - write in full

We have changed the acronym HF by the words heart failure through the text. 

Material and Methods

2. Provide details of psychometrics of two tools

Now we stand:

The European Heart Failure Self-Care Behavior Scale (EHFScBS) is a self-administered questionnaire consisting of 12 items on self-care skills and attitudes that allow monitoring of the health-related behavior of heart failure patients [28]. Each item is rated on a scale of 5 points, from 1 “I don’t agree at all” to 5 “I completely agree”. Scores range from 12 to 60 and lower scores reflect better self-care. The content validity of the EHFScBS was verified by concept analysis, expert panels and face-validity. Psychometric properties were validated using concurrent validity assessment, and internal consistency showed a Cronbachs’ alpha of 0.81. The EHFScBS has been used widely in research as it is a unique instrument and has been adapted to Swedish, Dutch, Finnish, Italian, Turkish and Spanish.

Psychometric properties were validated using concurrent validity assessment, and internal consistency showed a Cronbachs’ alpha of 0.81. The EHFScBS has been used widely in research as it is a unique instrument and has been adapted to Swedish, Dutch, Finnish, Italian, Turkish and Spanish.

Health literacy was measured using the HLS-EU-Q47 questionnaire, which includes 47 items related to the management of health information, and aspects of healthcare, disease prevention and health promotion. The HLS-EU-Q47 in Spanish has  unifactorial structure with a hish internal consistency (Cronbach's alpha:=0.97). The questionnaire provides a general index of health literacy, and the authors defined four levels of health literacy in the general public: inadequate, problematic, sufficient, and excellent.

3. NYHA - please write in full first time used

We have written the NYHA in full the first time, now we stand:

Other clinical characteristics collected were: severity of the heart failure, assessed with the New York Heart Association scale (NYHA).

4. Specify groups which will be compared in data analysis section

Now we stand:

Between-group differences in overall self-care were tests by ANOVA for the overall person mean location by groups of: sex, age (groups according quartiles), marital status, cohabitation, academic level, NYHA classification, adherence to drug treatment, polypharmacy, hospital admission last year and Health literacy index (Inadequate, problematic, sufficient, excellent).

Results.

5. Please report data consistently eg. mean (sd) and n (%)

Now data is reported consistently through all the text.

6. I would suggest doing Table 1 by Gender.

We have reported the number by gender. Now in table 1 we stand:

N=318

Females

Males

Sex. n   (%)

163 (51.26%)

155 (48.74%)

Age. Mean   (SD)

77.96 (8.67)

78.85 (7.68)

75.97 (9.23)

Marital   status. n (%)

 Single

25 (7.86%)

13 (7.98%)

12 (7.74%)

 Married or living together

180 (56.60%)

59 (36.20%)

121 (78.06)

 Divorced

4 (1.26%)

1 (0.61%)

3 (1.94%)

 Widowed

109 (34.28%)

90 (55.21%)

19 (12.26%)

Cohabitation.   n (%)

 Living alone

79 (24.84%)

62 (38.04%)

17 (10.97%)

 2

185 (58.18%)

79 (48.47%)

106 (68.39%)

 >2

54 (16.98%)

22 (13.50%)

32 (20.65%)

Sources   of household income. n (%)

 Employment / unemployment allowance

15 (4.72%)

6 (3.68%)

9 (5.81%)

 Disability pension

21 (6.60%)

3 (1.84%)

18 (11.61%)

 Retirement pension

174 (54.72%)

51 (31.29%)

123 (79.35%)

 Widow's pension

64 (20.13%)

62 (38.04%)

2 (1.29%)

 Social aid   (welfare financial benefits)

14 (4.40%)

13 (7.98%)

1 (0.65%

Academic   level. n (%)

 Primary education not completed

185 (58.18%)

117 (71.78%)

68 (43.87%)

 Primary education

76 (23.89%)

34 (20.86%)

42 (27.10%)

 Secondary Education / Vocational studies.

52 (16.35%)

12 (7.36%)

40 (25.81%)

 University degree or higher.

5 (1.57%)

0 (0.00%)

5 (3.23%)

NYHA. n   (%)

 I

88 (27.67%)

43 (26.38%)

45 (26.38%)

 II

163 (51. 25%)

83 (50.92%)

80 (51.61%)

 III

64 (20.13%)

35 (21.47%)

29 (18.71%)

 IV

3 (0.94%)

2 (1.23%)

1 (0.65%)

Non-adherence   to drug treatment. n (%)

240 (75.5%)

125 (76.69%)

115 (74.19%)

Health   Literacy Index (min: 0, max: 50). Mean (SD)

25.44 (9.05

23.44 (9.22)

27.54 (8.41)

Visits to   PHC nurse in last year. Mean (SD)

12.48 (11.02)

13.77 (11.90)

11.22 (9.90)

Visits to   PHC family physician in last year. Mean (SD)

10.19 (6.30)

10.77 (11.90)

9.59 (5.75)

7. Include in Table 1 the total possible score for HLI etc... to provide context for reader.

We have added the information about the results in the text and in the table 1 se review #6.

8. Table 2 - specify details of items.

Now we stand:

Item

Category frequency (% of responses)

1

2

3

4

5

I don’t agree at all

I completely agree

1. I weigh myself every day.

129   (40.6%)

63   (19.8%)

50   (15.7%)

50   (15.7%)

26   (8.2%)

2. If I get short of breath, I take it easy.

23   (7.2%)

15   (4.7%)

23   (7.2%)

34   (10.7%)

223   (70.1%)

3. If my shortness of breath increases, I   contact my doctor or nurse.

45   (14.2%)

25   (7.9%)

38   (11.9%)

47   (14.8%)

163   (51.3%)

4. If my feet/legs become more swollen than   usual, I contact my doctor or nurse.

45   (14.2%)

34   (10.7%)

37   (11.6%)

64   (20.1%)

138   (43.4%)

5. If I gain 2 kg in 1 week, I contact my   doctor or nurse.

128   (40.3%)

50   (15.7%)

39   (12.3%)

37   (11.6%)

64   (20.1%)

6. I limit the amount of fluids I drink (not   more than 1.5–2 l/day).

59   (18.6%)

36   (11.3%)

51   (16.0%)

61   (19.2%)

111   (34.9%)

7. I take a rest during the day.

9   (2.8%)

8   (2.5%)

14   (4.4%)

41   (12.9%)

246   (77.4%)

8. If I experience increased fatigue, I   contact my doctor or nurse.

51   (16.0%)

36   (11.3%)

39   (12.3%)

55   (17.3%)

137   (43.1%)

9. I eat a low salt diet.

22 (6.9%)

18   (5.7%)

35   (11.0%)

64   (20.1%)

179   (56.3%)

10. I take my medication as prescribed.

4   (1.3%)

0   (0.0%)

4   (1.3%)

11   (3.5%)

299   (94.0%)

11. I get a flu shot every year.

48   (15.1%)

8   (2.5%)

11   (3.5%)

9   (2.8%)

242   (76.1%)

12. I exercise regularly. Females

63   (19.8%)

34   (10.7%)

67   (21.1%)

46   (14.5%)

108   (34.0%)

9. Section 3.3 - It would have been good to see the table of OR (CI) in a table

We have added the table 5 with the OR and the 95% CI. We have clarified this analysis in the methods section. Now we stand:

Methods section.

In groups with significant differences we analyzed the mean difference with the Bonferroni pairwise comparison test, results logits measures were transformed in odds ratio (OR) with the 95% CI.

Results section:

The pairwise comparison of these groups showed in table 5 showed that the OR of adherence to self-care in the group with secondary education or higher compared with patients who had not completed primary education was protective; and, adherence to self-care was also protective in the group with sufficient health literacy compared to the group with inadequate health literacy.

Table 5. Pairwise comparison of the means (Bonferroni method). The logit measure is expressed in OR. * is the reference group.

Group

Sub-group

OR (95% CI)

Academic level

Primary education not completed*

Primary education completed

1.15   (0.74 to 1.78)

Secondary education or higher

0.60   (0.37 to 0.97)

Health literacy Index

Inadequate*

Problematic

0.70   (0.45 to 1.09)

Sufficient

0.35   (0.21 to 0.60)

Excellent

0.52   (0.15 to 1.83)

10. Why was correlation of PHC visit Nurse and PHC physician visit done?

The hypothesis was that the number of visits to health care professionals correlates positively with the adherence to self-care. We expected that patient with more visits to health care professional, and particularly to nurses, would have higher adherence to selfcare.

11. Please add the psychometrics (Chronbach Alpha) as stated in methodology in the results.

Now we stand in the methods section:

After splitting the item, the separation index (analogous to Cronbach alpha) was 0.99 indicating an optimal reliability. 

Reviewer 2 Report

Overall, this is a solid paper that examines the self-care practices of HF patients in primary health care. There are suggested edits throughout that would make the paper stronger and more cohesive. I do think the discussion needs a bit more work in that it seems to drift off into other areas (incorporating different measures, program development) without any context.

Introduction:

Page 1, Line 43: change to "HF tends to worsen over time, and hospital admissions are..."

Page 1, Line 44: change to "Around 26 million adults have HF worldwide."

Page 2, Line 1: add comma after "populations"

Page 2, Line 14: Can you clarify? Does PHC planned care only occur after admission? What if pts are diagnosed with HF outpatient? How does this process work?

Page 2, Line 15: Change "permit" wording. Unclear.

Page 2, Line 18: Take out comma after "HF"

Page 2, Lines 21-22: How is this sentence different from the one earlier (lines 11-12)? They seem to be conveying the same thing.

Page 2, Line 32: change "are" to "is"

Page 2, Line 33: change wording "have been made". Unclear. Maybe "have examined"?

Page 2, Line 34: Take out comma after "self-care"

Methods:

Page 2, Line 42: Missing word "patient" (in PHC patients diagnosed with...)

Page 3, Lines 16-18: No edits needed. I just wanted to comment that this is wonderful protocol!

Page 3, Line 23: I would change to say "higher scores reflect poorer self-care." It makes it easier to read and understand since most people are used to looking at higher scores for effects.

Results:

Page 5, Line 1: change wording to "the majority were NYHA II (51.3%)"

Page 5, Lines 3-4: Can you specify what the mean age of males vs females was?

Table 1: You have it formatted where there is a row for "table 1 (Cont)", yet it appears that is not necessary given the formatting. The entire table appears as one graphic.

Table 2: Confusing. What exactly are the items being represented? Also, there is a distinction for bolded items, but it does not appear that any are bolded. This table needs some revision.

Line 10: Place comma before quotation marks ("prescribed,")

Discussion:

Line 15: Associated with better adherence? It's assumed, but I would specify.

Second paragraph re: changing to dichotomous seems to be out of context with purpose of paper. This should either be identified as a secondary aim of the paper or used for another paper altogether. It seems out of place.

Line 17: Can you clarify how they change their behavior according to physical functioning? This is vague and unclear.

Lines 21-22: Providers who work with HF pts know that daily weighing is necessary and important. Can you say more about this? It seems like this sentence does not add much given that basic knowledge that most already have about HF self-care needs.

Line 33: Take out comma after "Our results suggest that..."

Again, the topic of creating criteria to benchmark practices seems somewhat out of place considering the aim of the study. I would rework aims of paper to include this.

Author Response

Reviewer 2

Introduction:

1. Page 1, Line 43: change to "HF tends to worsen over time, and hospital admissions are..."

Now we stand:

HF is considered stable when symptoms and signs are unchanged for ≥ 1 month. HF tends to worsen and hospital admissions are frequent [1].

2. Page 1, Line 44: change to "Around 26 million adults have HF worldwide."

Now we stand:

Around 26 million adults have HF worldwide [3]

3.  Page 2, Line 1: add comma after "populations"

Now we stand:

In countries with aging populations, HF is becoming…

4. Page 2, Line 14: Can you clarify? Does PHC planned care only occur after admission? What if patients are diagnosed with HF outpatient? How does this process work?

We thank the review to clarify, now we stand:

When patients are admitted to hospital, the PHC receives an automatic warning, and before the discharge the hospital send a report to the PHC to the nurse and the family physician. They carry out a phone call to appoint a visit at home or at the PHC center [9,10]. While patient care in the PHC system is lifelong, patients with a recent hospital admission receive higher attention to set a new plan of care.

5. Page 2, Line 15: Change "permit" wording. Unclear.

Now we stand:

PHC clinicians’ objectives in HF patients includes activities of self-management to improve patient decision-making, such as education on self-detection of symptoms if HF worsens (e.g. weight control to detect fluid retention), control of adherence to prescribed drugs, diet and exercise routines, and symptom management [11,12].

6. Page 2, Line 18: Take out comma after "HF"

Now we stand:

Clinical guidelines on HF recommend drawing up follow-up procedures and standardizing interventions.

7. Page 2, Lines 21-22: How is this sentence different from the one earlier (lines 11-12)? They seem to be conveying the same thing.

We thank the review; we have removed the sentence to avoid redundancy.

8. Page 2, Line 32: change "are" to "is"

Now we stand,

Self-care follow up is associated with a better health status in HF patients [21,22].

9. Page 2, Line 33: change wording "have been made". Unclear. Maybe "have examined"?

Now we stand:

Most studies have examined in patients with unstable or acute HF attended in hospitals or hospital outpatient clinics during the clinical follow-up [23]

10. Page 2, Line 34: Take out comma after "self-care".

We have removed the comma.

Methods:

11. Page 2, Line 42: Missing word "patient" (in PHC patients diagnosed with...)

Now we stand:

… in PHC patients diagnosed with chronic HF, and to examine potential relationships between self-care practices and sociodemographic and clinical characteristics.

12. Page 3, Lines 16-18: No edits needed. I just wanted to comment that this is wonderful protocol!

We appreciate this comment!

13. Page 3, Line 23: I would change to say "higher scores reflect poorer self-care." It makes it easier to read and understand since most people are used to looking at higher scores for effects.

Now we stand:

Scores range from 12 to 60 and higher scores reflect poorer self-care.

Results:

14. Page 5, Line 1: change wording to "the majority were NYHA II (51.3%)"

Now we stand:

the majority were  NYHA II (51.3%)

15. Page 5, Lines 3-4: Can you specify what the mean age of males vs females was?

Now we stand:

The mean age of women was significantly higher (mean age of woman=79.85 vs mean age of males=75.97, difference 3.88 years, t=-4.086; P<0.001)

16. Table 1: You have it formatted where there is a row for "table 1 (Cont)", yet it appears that is not necessary given the formatting. The entire table appears as one graphic.

We thank this review. We have deleted the with the text table 1 (Cont). Now we stand in table 1:

N=318

Females

Males

Sex. n (%)

163 (51.26%)

155 (48.74%)

Age. Mean (SD)

77.96 (8.67)

78.85 (7.68)

75.97 (9.23)

Marital status. n (%)

 Single

25 (7.86%)

13 (7.98%)

12 (7.74%)

 Married or living together

180 (56.60%)

59 (36.20%)

121 (78.06)

 Divorced

4 (1.26%)

1 (0.61%)

3 (1.94%)

 Widowed

109 (34.28%)

90 (55.21%)

19 (12.26%)

Cohabitation. n (%)

 Living alone

79 (24.84%)

62 (38.04%)

17 (10.97%)

 2

185 (58.18%)

79 (48.47%)

106 (68.39%)

 >2

54 (16.98%)

22 (13.50%)

32 (20.65%)

Sources of household   income. n (%)

 Employment / unemployment allowance

15 (4.72%)

6 (3.68%)

9 (5.81%)

 Disability pension

21 (6.60%)

3 (1.84%)

18 (11.61%)

 Retirement pension

174 (54.72%)

51 (31.29%)

123 (79.35%)

 Widow's pension

64 (20.13%)

62 (38.04%)

2 (1.29%)

 Social aid (welfare financial benefits)

14 (4.40%)

13 (7.98%)

1 (0.65%

Academic level. n (%)

 Primary education not completed

185 (58.18%)

117 (71.78%)

68 (43.87%)

 Primary education

76 (23.89%)

34 (20.86%)

42 (27.10%)

 Secondary Education / Vocational studies.

52 (16.35%)

12 (7.36%)

40 (25.81%)

 University degree or higher.

5 (1.57%)

0 (0.00%)

5 (3.23%)

NYHA. n (%)

 I

88 (27.67%)

43 (26.38%)

45 (26.38%)

 II

163 (51. 25%)

83 (50.92%)

80 (51.61%)

 III

64 (20.13%)

35 (21.47%)

29 (18.71%)

 IV

3 (0.94%)

2 (1.23%)

1 (0.65%)

Non-adherence to drug   treatment. n (%)

240 (75.5%)

125 (76.69%)

115 (74.19%)

Health Literacy Index   (min: 0, max: 50). Mean (SD)

25.44 (9.05

23.44 (9.22)

27.54 (8.41)

Visits to PHC nurse in   last year. Mean (SD)

12.48 (11.02)

13.77 (11.90)

11.22 (9.90)

Visits to PHC family   physician in last year. Mean (SD)

10.19 (6.30)

10.77 (11.90)

9.59 (5.75)

17. Table 2: Confusing. What exactly are the items being represented? Also, there is a distinction for bolded items, but it does not appear that any are bolded. This table needs some revision.

We thank this review. Table 2 proves that the data did not fit the Rasch first point of the Rasch quality criteria Explained in Page 4, lines: 4-13: 1) Each category should contain ≥ 10 observations. 2) Average measures must be ordered. These criteria is needed to verify that categories contribute with substantive meaning to the rating scale functioning. We have added some information to clarify the table. Now we stand:

Table 2. Analysis of the rating scale structure of the EHFScBS questionnaire. Items and frequency (%) of answers in each category. Rasch analysis requires categories with ≥ 10 observations. (In bold, categories with <10 of responses).

Item

Category frequency   (% of responses)

1

2

3

4

5

I don’t agree at   all

I completely agree

1. I weigh   myself every day.

129 (40.6%)

63 (19.8%)

50 (15.7%)

50 (15.7%)

26 (8.2%)

2. If I get   short of breath, I take it easy.

23 (7.2%)

15 (4.7%)

23 (7.2%)

34 (10.7%)

223 (70.1%)

3. If my   shortness of breath increases, I contact my doctor or nurse.

45 (14.2%)

25 (7.9%)

38 (11.9%)

47 (14.8%)

163 (51.3%)

4. If my   feet/legs become more swollen than usual, I contact my doctor or nurse.

45 (14.2%)

34 (10.7%)

37 (11.6%)

64 (20.1%)

138 (43.4%)

5. If I   gain 2 kg in 1 week, I contact my doctor or nurse.

128 (40.3%)

50 (15.7%)

39 (12.3%)

37 (11.6%)

64 (20.1%)

6. I limit   the amount of fluids I drink (not more than 1.5–2 l/day).

59 (18.6%)

36 (11.3%)

51 (16.0%)

61 (19.2%)

111 (34.9%)

7. I take a rest during the   day.

9 (2.8%)

8 (2.5%)

14 (4.4%)

41 (12.9%)

246 (77.4%)

8. If I   experience increased fatigue, I contact my doctor or nurse.

51 (16.0%)

36 (11.3%)

39 (12.3%)

55 (17.3%)

137 (43.1%)

9. I eat a   low salt diet.

22 (6.9%)

18 (5.7%)

35 (11.0%)

64 (20.1%)

179 (56.3%)

10. I take my medication as   prescribed.

4 (1.3%)

0 (0.0%)

4 (1.3%)

11 (3.5%)

299 (94.0%)

11. I get a flu shot every   year.

48 (15.1%)

8 (2.5%)

11 (3.5%)

9 (2.8%)

242 (76.1%)

12. I   exercise regularly. Females

63 (19.8%)

34 (10.7%)

67 (21.1%)

46 (14.5%)

108 (34.0%)

18. Line 10: Place comma before quotation marks ("prescribed,")

Now we stand:

The item with the highest likelihood of endorsement was, “I take my medication as prescribed”.

Discussion:

19. Line 15: Associated with better adherence? It's assumed, but I would specify.

Now we stand:

Factors such as educational level and health literacy were associated with better adherence to treatment.

20. Second paragraph re: changing to dichotomous seems to be out of context with purpose of paper. This should either be identified as a secondary aim of the paper or used for another paper altogether. It seems out of place.

We thank the review. We used Rasch analysis as a method to examine the particular health-care behaviors identified in the items of the EHFScBS questionnaire. We did not plan a methodological study to scrutinize the psychometrics of the questionnaire as it would have required an heterogenous sample. However, as to our knowledge this was the first time that the EHFScBS has been studied with a Rasch analysis we have considered interesting to add a paragraph explaining the findings. Therefore, now we stand:

The Rasch analysis of the goodness-of-fit of data to the Rasch model of the EHFScBS did not changed the questionnaire itself, but provided insights into the rating scale functioning of the questionnaire. The sample of patients answered the questionnaire in a clear dichotomous pattern, suggesting the 5-category rating scale could be changed to a simple Yes or No answer. To our knowledge neither the original validation study or any subsequent studies using the EHFScBS have commented on this aspect [21,27,28,35,36]. As far as we know, this is the first analysis of the EHFScBS using a Rasch model; all other studies used classical test theory methods where the measure of self-care is the sum of scores, and therefore more categories of the rating scale are useful but they add more variability. However, using the Rasch model avoids this issue, and also provides researchers results at item level [37]. We believe that Rasch model analysis might help clinicians to monitor patient attitudes, and the results could allow physicians and family nurses to answer the question “what should we do next?” more accurately, as well as personalize healthcare plans with questionnaires used in daily practice [38,39].

21. Line 17: Can you clarify how they change their behavior according to physical functioning? This is vague and unclear.

We thank the review. Now we stand:

Other behaviors with high endorsement were related to resting when feeling fatigue symptoms (e.g.: “I take a rest during the day” or “If I am short of breath, I take it easy”). However, we think that these behaviors are not strange because physical impairment impels patients to change attitudes toward any issue or daily activity.

22. Lines 21-22: Providers who work with HF patients know that daily weighing is necessary and important. Can you say more about this? It seems like this sentence does not add much given that basic knowledge that most already have about HF self-care needs.

Now we state:

Items with lower endorsement were related to daily weighing and contacting clinicians over increases in body weight. Weighting is strongly recommended in heart failure guidelines, because of is used to change the dosage of diuretics and alerts physicians to sudden worsening in the health status [1,11]. Other studies have showed low adherence to this basic self-care behavior [27,36], this might be regarded as a process of adaptation: patients may tolerate edema, weight gain, and fatigue for as long as seven days before seeking healthcare attention [44]. We think that evidence of low  adherence to weighing might alert to health care professionals to implement new strategies to help patients.

23. Line 33: Take out comma after "Our results suggest that..."

We have deleted this review following the advice of review #24.

24. Again, the topic of creating criteria to benchmark practices seems somewhat out of place considering the aim of the study. I would rework aims of paper to include this.

We thank this review, now we state:

Our results suggest health care providers might seek , effective measures to help PHC professionals to follow up patients with heart failure.
